# Characterization of Effects of Different Tea Harvesting Seasons on Quality Components, Color and Sensory Quality of “Yinghong 9” and “Huangyu” Large-Leaf-Variety Black Tea

**DOI:** 10.3390/molecules27248720

**Published:** 2022-12-09

**Authors:** Fei Ye, Xinbo Guo, Bo Li, Haiqiang Chen, Xiaoyan Qiao

**Affiliations:** 1Tea Research Institute, Guangdong Academy of Agricultural Sciences, Guangdong Provincial Key Laboratory of Tea Plant Resources Innovation and Utilization, Guangzhou 510640, China; 2Institute of Fruit and Tea, Hubei Academy of Agricultural Sciences, Wuhan 430064, China; 3Tuguanya Agricultural Technology Extension Center, Danjiangkou 442700, China

**Keywords:** harvesting seasons, large leaf variety black tea, catechins, caffeine, color, physicochemical qualities

## Abstract

Harvesting seasons are crucial for the physicochemical qualities of large-leaf-variety black tea. To investigate the effect of harvesting seasons on physicochemical qualities, the color and sensory characteristics of black tea produced from “Yinghong 9” (Yh) and its mutant “Huangyu” (Hy) leaves were analyzed. The results demonstrated that Hy had better chemical qualities and sensory characteristics, on average, such as a higher content of tea polyphenols, free amino acids, caffeine, galloylated catechins (GaCs) and non-galloylated catechins (NGaCs), while the hue of the tea brew (ΔE*ab and Δb*) increased, which meant that the tea brew was yellower and redder. Moreover, the data showed that the physicochemical qualities of SpHy (Hy processed in spring) were superior to those of SuHy (Hy processed in summer) and AuHy (Hy processed in autumn), and 92.6% of the total variance in PCA score plots effectively explained the separation of the physicochemical qualities of Yh and Hy processed in different harvesting seasons. In summary, Hy processed in spring was superior in its physicochemical qualities. The current results will provide scientific guidance for the production of high-quality large-leaf-variety black tea in South China.

## 1. Introduction

Tea (*Camellia sinensis* L.) is one of the most traditional and popular beverages consumed worldwide due to its physicochemical qualities, health benefits and sensory flavors [1,2]. Variations in the physicochemical qualities of black tea are directly related to organoleptic properties and health benefits [3,4]. The variety of tea, harvesting season and other factors, including polyphenols (catechins), caffeine, free amino acids, and even theaflavins (TFs) and thearubigins (TRs), contribute to the physicochemical qualities of black tea [5,6,7,8,9]. Previous research revealed that the catechin profile, as a substrate of TFs and TRs, was identified as one of the seasonal discriminating parameters of green tea and fresh leaves [10,11,12].

Etiolated tea refers to special germplasm growing white leaves or yellowish leaves. The chemical composition attributes of etiolated tea leaves are sensitive to environmental factors, such as lower temperature and light intensity [13,14]. The mutant “Huangyu” is derived from the cultivar “Yinghong 9” (*Camellia sinensis* var. *assamica*), which is a large-leaf tea plant grown in South China. The shoot color of the mutant is yellow throughout the year under high light intensity. Many studies have been carried out on the mechanism of leaf color formation using transcriptomic, proteomic and metabolomic analyses [15,16,17]. Previous studies have also highlighted the remarkable differences in antioxidant properties and sensory quality between the mutant “Huangyu” and “Yinghong 9” [18,19]. There is no doubt that harvesting seasons may have a certain influence on the bioactive components, color and sensory characteristics of “Huangyu” and “Yinghong 9” large-leaf-variety black tea. However, there is little information about the effect of harvesting seasons on large-leaf-variety black tea characteristics.

Based on the above analysis, a comprehensive investigation is needed to explore the tea compound metabolic profile of large-leaf-variety black tea from different seasons. Therefore, this study aimed to explore the effect of harvesting seasons (spring: 5 April; summer: 10 July; and autumn: 12 September) on the physicochemical qualities and sensory characteristics of black tea produced by “Yinghong 9” and “Huangyu” leaves. These findings could provide guidance for the production of high-quality large-leaf-variety black tea in South China and other black-tea-producing areas.

## 2. Results and Discussion

### 2.1. Effect of Different Large-Leaf Varieties on Black Tea Physicochemical Qualities

The physicochemical qualities of different large-leaf varieties, Yh and Hy, are shown in Figure 1. The contents of total tea polyphenols (Hy, 13.95%; Yh, 11.82%), free amino acids (Hy, 3.24%; Yh, 3.08%) and caffeine (Hy, 3.64%; Yh, 2.89%) were significantly higher in Hy compared to those in Yh (*p* < 0.01), while the content of soluble sugar (Hy, 6.33%; Yh, 7.84%) markedly decreased in Hy compared to that in Yh (*p* < 0.01) (Figure 1A). Black tea has many functional properties, such as anticarcinogenic, antioxidation, hypolipidemic and antimicrobial activities, on account of tea polyphenols and amino acids, which are the main nutrient components in tea [1,20]. Previous studies reported that the amino acid content in etiolated leaves was higher due to the degradation of chloroplast structural proteins induced by blocked photosynthetic pigments [13,14,21,22,23,24]. Caffeine is the principal alkaloid in tea leaves, responsible for imparting a bitter taste in tea infusion. Caffeine tastes bitter, while the more amino acids, the more umami and sweetness in the tea infusion [25]; the combination of these two chemical components makes the Hy tea infusion mellow and adds depth to the taste. The results of this evaluation are consistent with a previous report [14], which indicated that the increased content of caffeine in mutant Hy was not affected by the deficiency of chlorophyll, which is probably because of the genotype of Hy. 

The content of residual catechins in Yh was much higher than that in Hy (Figure 1B). The amount of EGC after oxidation was dominant in both Yh and Hy; in addition, the content of EGC was significantly higher in Hy (13.20 mg·g^−1^) than in Yh (9.40 mg·g^−1^). The principal galloylated catechins (GaCs) are EGCG, GCG and ECG, while EGC, C and EC constitute the non-galloylated catechins (NGaCs). The contents of GaCs and NGaCs in Hy (5.91 mg·g^−1^, 18.82 mg·g^−1^) were approximately 2.64- and 1.43-fold higher than those in Yh (2.24 mg·g^−1^, 13.16 mg·g^−1^), respectively. Thus, we speculated that the characteristics of the large-leaf varieties were responsible for the variation in catechins throughout the harvesting period. Therefore, our finding is consistent with a previous study [26].

The principal substrates of TFs are EGC and EGCG, and up to 75% of catechin substrates may ultimately transform into TRs [27]. To further explore the impact of different large-leaf varieties on black tea compounds, the individual catechin oxidation efficiency was analyzed. The oxidation efficiencies of ECG, GCG, EGC, C, EC, GaCs and NGaCs in Yh were much lower compared to those in Hy (Figure 1C). The above results could explain why the final TF, TR and TB contents in Yh were significantly different from those in Hy (Figure 1D). Upon further analysis, the oxidation efficiency is related to polyphenol oxidase (PPO) and peroxidase (POD) enzymes, which play a rate-limiting role in oxidation. Previous studies revealed that PPO activity increased [28], while the POD activity decreased [29] in etiolated tea leaves, which provides rich and complex TF, TR and TB content differences between Yh and Hy. 

### 2.2. Effect of Large-Leaf Variety on Black Tea Sensory Characteristics

The natural performance and radar chart of sensory analysis intuitively display the characteristics of Yh and Hy, as shown in Figure 2. The natural performance of Hy caused it to become yellower, and the infusion tasted less brisk, more bitter and more mellow (Figure 2A,B) due to the higher levels of NGaCs (1.43-fold) and caffeine (1.26-fold) in Hy. NGaCs showed a stronger taste of astringency than GaCs [30], while caffeine enhanced the tea’s bitterness [25,30]. On the other hand, the tea infusion of Hy tasted more mellow and bitter than that of Yh. 

The color difference (ΔE*ab) was used to differentiate tea infusions, orange juice and wooden surfaces [31,32,33]. ΔE*ab and Δb* of Hy were significantly higher than those of Yh, and the Δa* of Hy was also lower than that of Yh (Figure 2C), suggesting that the infusion color of Hy was more yellow and red; these data are in agreement with the sensory analysis (Figure 2A,B). Based on the results of the physicochemical qualities, the black tea made from Hy growing yellowish leaves contains an abundance of caffeine, free amino acids, GaCs and NGaCs. The different oxidation rates of NGaCs contribute to different contents of TRs and TFs, leading to remarkable differences in the briskness and infusion color of black tea. 

### 2.3. Effect of Different Harvesting Seasons on Physicochemical Qualities

The results of the physicochemical quality analysis of Yh and Hy over three harvesting seasons (spring, summer and autumn) are presented in Table 1. The content of tea polyphenols significantly varied among the three harvesting seasons, but the peak content of tea polyphenols in black tea was in summer, regardless of the variety. This is because tea leaves accumulate catechins to protect themselves against damage from ultraviolet rays [34,35]. The contents of soluble sugar and caffeine increased significantly (*p* < 0.01), while the content of free amino acids markedly decreased in black tea processed from spring to summer; the same variation in the compound contents was observed in both Hy and Yh (Table 1), and the results are consistent with previous studies [10,36]. Much research revealed that caffeine synthesis increased in response to increased light intensity; however, amino acids were contrary to caffeine [37,38,39]. The content of sugar in Hy in summer and autumn decreased by 25.29% and 19.19% compared with that in Yh, respectively (Table 1). The contents of TFs and TRs remarkably decreased in summer and autumn, while the ratio of TFs/TRs increased for both Hy and Yh.

Based on the above information, we speculated that the decrease in the content of sugar in Hy was due to seasonal variations in light intensity, resulting in reduced sugar biosynthesis [40]. The same result was revealed in Kangra tea, where variation in harvesting seasons led to changes in TF content and infusion color [10,41]. 

In this study, there was a remarkable accumulation of GaC and NGaCs in black tea processed in summer or autumn for both Hy or Yh (Figure 3A). The contents of GaCs and NGaCs were the highest in summer, indicating more bitterness and briskness in SuHy and Suhy. Therefore, our findings are consistent with a previous study [34]. The ratio of GaCs/NGaCs in Yh and Hy significantly increased in summer and autumn compared to that in spring (Figure 3B), and the seasonal distribution pattern of the ratio appeared to be different between Yh and Hy. The data suggested that different photoprotection requirements resulted in variations in the GaC/NGaC ratio [34,42]. In addition, the content ratio of GaCs/NGaCs was also used to monitor seasonal changes in the tea shoots of the *assamica* variety grown on Australian tea farms [12]. Therefore, the significantly different distributions of the ratio of GaCs/NGaCs could be used as a parameter for monitoring the physicochemical qualities of black tea.

For a better overview of the effect of different harvesting seasons on physicochemical qualities, the PCA method was performed to classify the performance and select informative variables in terms of computational efficiency [43]. PCA was used to isolate the variables responsible for differences among the three seasonal harvests. As shown in Figure 4B, 92.6% of the total variance (69.3%, 13.5% and 9.8%, respectively) in the data was described. Obviously, the clear separation of Yh and Hy based on PC1 and PC2 (Figure 4A) indicated the significant effects of different seasonal harvests on the biochemical components of black tea. SpYh, SpHy, AuYh, AuHy, SuYh and SuHy were separated (Figure 4B).

Sensory evaluation displayed differential characteristics over the three harvest seasons, as shown in Figure 4C. SpYh had good quality and tasted more brisk because of the low contents of caffeine, GaCs and NGaCs compared with SuYh and AuYh. The quality of SpHy was also superior to that of SuHy and AuHy, and an obvious bitter taste was noted for SuHy, which contained the highest amounts of caffeine and NGaCs. ∆Eab* significantly decreased in Hy from spring to autumn but increased in Yh, as shown in Figure 4D, demonstrating that the infusion color was much more impacted by harvesting seasons, consistent with the report that the total color and brightness of Kangra orthodox tea was associated with weather conditions [41].

Overall, these results demonstrated that harvesting seasons affected the sensory quality of Yh and Hy due to variations in the chemical qualities. The PCA model analysis results were remarkably similar to those of the sensory quality evaluation of Yh and Hy processed in different seasons. We have identified the ratios of TFs/TRs and GaCs/NGaCs as potential markers for the quality evaluation of Hy from three harvesting seasons.

## 3. Materials and Methods

### 3.1. Experimental Materials 

Leaf samples of “Yinghong 9” and “Huangyu” were collected from the Tea Research Institute, Guangdong Academy of Agricultural Sciences, Yingde, China (24°18′ N, 35°21′ E; 6 m above sea level). Fresh tea leaves were plucked from 5-year-old plants in three harvesting seasons (spring, summer and autumn).

### 3.2. Methods of Black Tea Processing

Large-leaf-variety black teas were made from two varieties and processed using the same manufacturing procedure. The manufacturing procedure was carried out as follows: 25 kg of fresh leaves (one bud with two leaves) was left to wither until the water content reached 55% at 25~27 °C and 65~70% relative humidity. Next, the withered leaves were subjected to rolling for 45 min twice. Then, the rolled leaves were fermented in an artificial climate box (RXZ-328A, Changzhou City Solid Germany Instrument Co., Ltd.; Changzhou, China) at 28 °C and 90% relative humidity with air flow. Finally, the final rolled leaves and the resulting collected leaves were dried at 120 °C for 20 min and 80 °C for 2 h. Samples from each cultivar in every season consisted of three biological replicates, and technical measurements were performed in triplicate for each replicate sample, which means the processed leaves were split into three sub-batches, and each sub-batch was processed accordingly. The final rolled leaves were treated as a control to calculate the oxidation efficiency of catechins and color differences in the tea infusion.

### 3.3. Amino Acid and Soluble Sugar Determination

The tea samples were milled into powder and passed through a 20-mesh sieve. A total of 3 g of tea powder was extracted with 450 mL of boiling distilled water for 45 min. Then, the decoction was filtered using qualitative filter paper and quickly cooled to room temperature for chemical composition analysis. The free amino acid content was determined by using the ninhydrin colorimetric method [44]. Soluble sugar was measured by using the anthrone colorimetric method [45]. 

### 3.4. Tea Pigment Determination

Theaflavins (TFs), thearubigins (TRs) and theabrownines (TBs) were measured as described previously with minor modifications [46]. Briefly, 20 mL of boiling water was added to a 0.4 g tea sample, boiled for 5 min and cooled at room temperature. Then, 10 mL infusions were pipetted into 10 mL of isobutylmethylketone. Next, 2 mL of the upper layer was pipetted into a test tube, followed by 4 mL of ethanol and 2 mL of Flavognost reagent. The contents were mixed, and the color was allowed to develop for 15 min.

### 3.5. Total Polyphenol, Catechin and Caffeine Determination

The total polyphenol content was determined using the Folin–Ciocalteu method [47], and catechins and caffeine were determined as described previously with some modifications [5]. HPLC was used to determine the contents of catechins and caffeine, and the parameters were set as follows: the separation was carried out on a ZORBAX Eclipse Plus C18 column (Agilent, Santa Clara, CA, USA; 4.6 mm × 250 mm; 5 µm particle size), which was maintained at 35 °C. The mobile phases were 0.5% (*v*/*v*) formic acid (eluent A) and acetonitrile (eluent B). The gradient program was as follows: 0–20 min, 9% to 16% A; 20–40 min, linear gradient to 9% B. The catechin content in the control was used as the reference value to calculate the oxidation efficiency of catechins. 

### 3.6. Descriptive Sensory Analysis

The tea sensory quality was assessed by five professional tea tasters from the Tea Research Institute, Guangdong Academy of Agricultural Sciences, Yingde, China. The main assessment process was undertaken as followed. Briefly, each tea sample (3.00 g) was infused with 150 mL of freshly boiled water in a special pot for 5 min, and then the tea infusion was filtered out into a special bowl to be evaluated. Four attributes of the tea infusion were evaluated, including the infusion’s redness, taste of briskness, taste of mellowness and taste of bitterness. The following scales were used to rank the levels of these 4 parameters in the tea infusion: imperceptible—0; moderate—3; and strong—5.

### 3.7. Color Evaluation of Tea Infusion

According to the concept of *L*, *a* and *b* in three-dimensional color space, *L* represents lightness, with 100 for white and 0 for black. The *a* value indicates redness when positive and greenness when negative. Similarly, *b* shows yellowness when positive and blueness when negative. The color evaluation was carried out using a chromameter tristimulus color analyzer (Datacolor CheckII, Datacolor, Lawrenceville, GA, USA), and color differences (∆Eab*) were calculated according to Buchelt and Wagenführ [33]. The respective color values of the control were used as reference values.

### 3.8. Statistical Analysis

All measurements were performed at least in triplicate, and data were expressed as mean ± standard error of the mean (SEM). Tukey’s multiple comparison test was used to determine differences between groups, and *p* < 0.05 was considered the threshold for significance. GraphPad Prism 8 was used to perform statistical analysis.

Principal component analysis (PCA) was conducted using MetaboAnalyst 5.0. (https://www.metaboanalyst.ca/MetaboAnalyst/faces/home.xhtml) (accessed on 9 June 2013). To achieve better results of PCA, the raw data were subjected to the following pretreatments: Normalization of the samples was carried out by dividing each variable by the median. Data transformation was applied using logarithmic transformation to decrease the differences in intensities of large and small values. Data scaling was performed by autoscaling. 

## 4. Conclusions

In the present study, the effects of different tea harvesting seasons on the physicochemical qualities, color and sensory quality of Yh and Hy large-leaf-variety black tea were investigated. In the sensory quality analysis, Hy exhibited stronger mellow and bitter tastes compared to Yh because of the abundance of caffeine, amino acids, GaCs and NGaCs. The significantly different oxidation rates of NGaCs contributed to remarkable differences in TFs, TBs and infusion color. The infusion taste and color of Hy processed in three harvesting seasons showed significant variations and were different from those of Yh. The ratios of TFs/TRs and GaCs/NGaCs could be used as quality parameters for monitoring seasonal variations, even for distinguishing Hy from Yh. In summary, Hy had better physicochemical qualities than Yh. Moreover, Hy processed in spring had the best physicochemical qualities, and the results could provide scientific guidance for the production of high-quality large-leaf-variety black tea in South China.

## Figures and Tables

**Figure 1 molecules-27-08720-f001:**
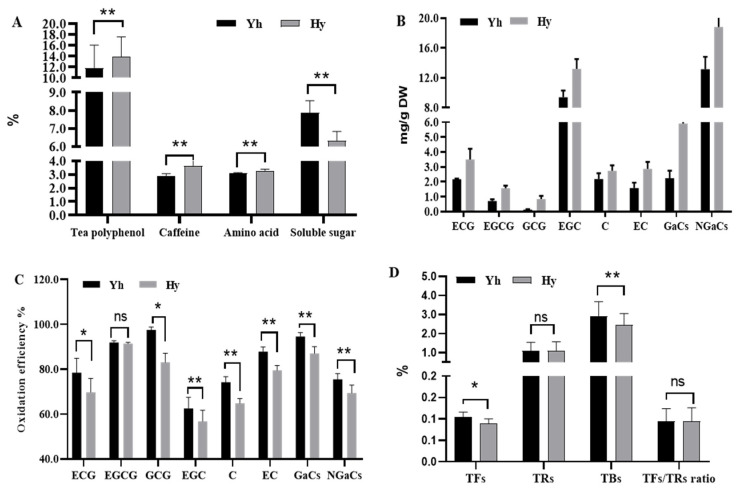
Analysis of physicochemical qualities. (**A**) The contents of tea polyphenols, caffeine, amino acids and soluble sugar in Yh and Hy; (**B**) the contents of ECG, EGCG, GCG, EGC, C, EC, GaCs and NGaCs in Yh and Hy; (**C**) catechin oxidation efficiency in Yh and Hy; (**D**) the contents of TFs, TRs and TBs and the TF/TR ratio in Yh and Hy. *, *p* < 0.05; **, *p* < 0.01; ns, not significant.

**Figure 2 molecules-27-08720-f002:**
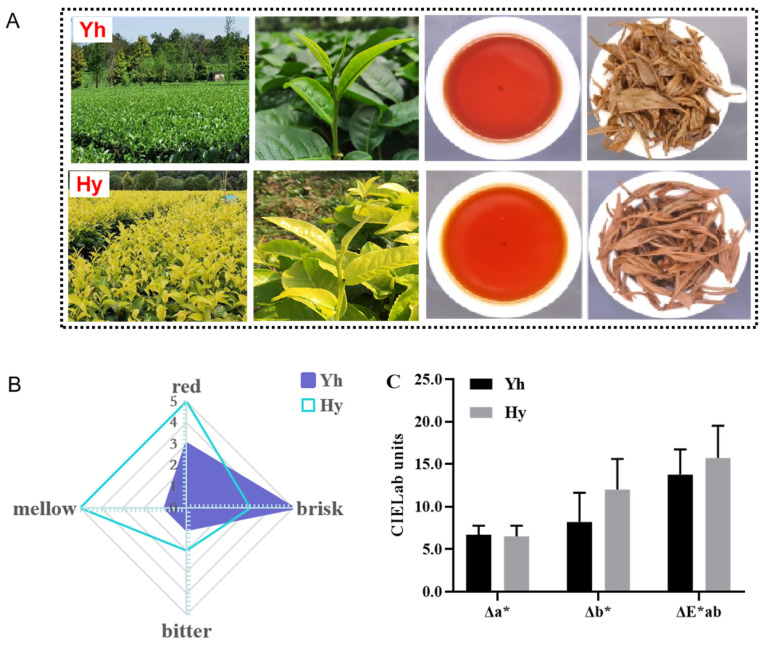
Sensory and color quality analysis of Yh and Hy. (**A**) Images of tea trees and tea infusions of Yh and Hy; (**B**) descriptive results of Yh and Hy shown in radar plot; (**C**) Δa*, Δb* and ΔE*ab values of Yh and Hy infusions.

**Figure 3 molecules-27-08720-f003:**
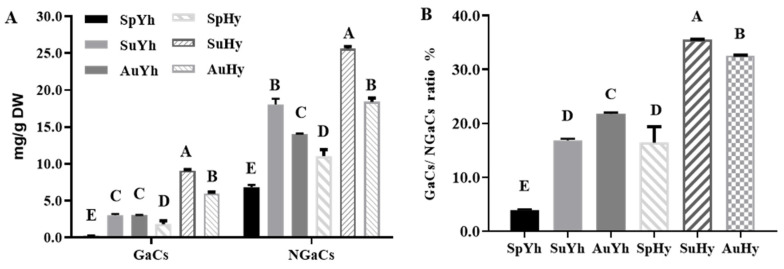
Analysis of GaCs and NGaCs in Yh and Hy over three harvesting seasons. (**A**) The contents of GaCs and NGaCs in Yh and Hy over three harvesting seasons; (**B**) the ratio of GaC/NGaCs in Yh and Hy over three harvesting seasons. Columns labeled with different capital letters represent significant differences (*p* < 0.01) between each other.

**Figure 4 molecules-27-08720-f004:**
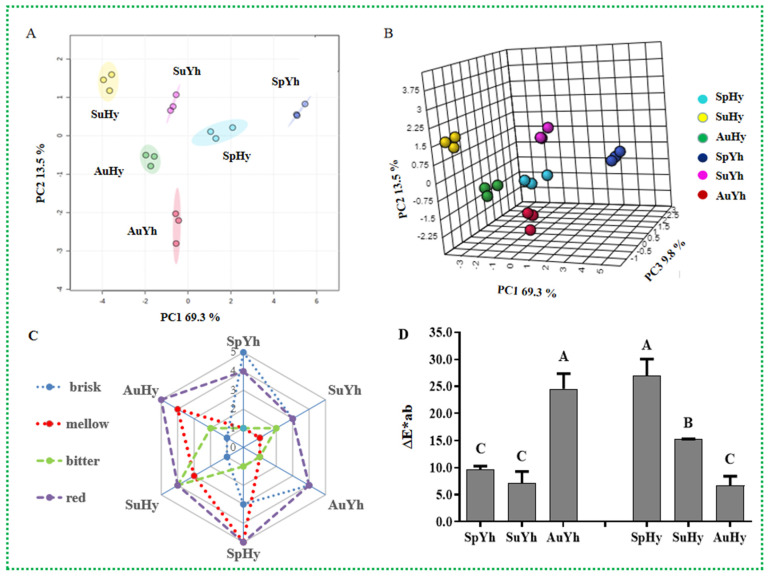
PCA and sensory and color analysis of Yh and Hy over three harvesting seasons. (**A**) Scatter plots of the first two principal components; (**B**) scatter plots of the first three principal components of PCA; (**C**) descriptive results of Yh and Hy over three harvesting seasons shown in radar plot; (**D**) the ΔE*ab values of tea infusions of Yh and Hy over three harvesting seasons.

**Table 1 molecules-27-08720-t001:** Physicochemical qualities of Yh and Hy processed in different harvesting seasons.

Parameters	Yh			Hy		
	Spring	Summer	Autumn	Spring	Summer	Autumn
Tea polyphenol/mg·g^−1^	64.30 ± 7.50 e	156.10 ± 11.30 b	134.40 ± 2.5 c	94.30 ± 8.40 d	174.70 ± 3.20 a	149.70 ± 8.80 b
Soluble sugar/mg·g^−1^	70.50 ± 2.50 c	84.90 ± 0.20 a	80.90 ± 1.50 b	57.80 ± 2.60 e	63.20 ± 0.50 d	69.00 ± 0.40 c
Caffeine/mg·g^−1^	27.20 ± 1.20 e	30.60 ± 1.10 d	31.40 ± 0.30 c	30.40 ± 0.40 d	40.50 ± 0.50 a	38.60 ± 0.90 b
Amino acids/mg·g^−1^	30.90 ± 0.60 b	30.20 ± 0.30 b	28.90 ± 0.20 b	34.10 ± 1.50 a	31.50 ± 0.80 b	31.70 ± 0.60 b
TBs/mg·g^−1^	38.90 ± 0.10 a	26.50 ± 0.60 c	22.10 ± 1.00 d	32.40 ± 0.30 b	19.60 ± 0.70 e	21.80 ± 0.70 d
TRs/mg·g^−1^	16.70 ± 0.60 a	9.40 ± 0.30 b	6.90 ± 0.20 d	17.20 ± 0.50 a	7.30 ± 0.60 d	8.40 ± 0.70 c
TFs/mg·g^−1^	1.00 ± 0.10 b	1.20 ± 0.10 a	0.70 ± 0.00 c	0.90 ± 0.10 b	0.90 ± 0.10 b	0.90 ± 0.20 b
TF/TR ratio	0.06 ± 0.10 c	0.13 ± 0.10 a	0.10 ± 0.00 b	0.05 ± 0.01 c	0.12 ± 0.01 a	0.11 ± 0.01 b

The same symbol within a column denotes no significant difference, whereas different symbols denote a significant difference (*p* < 0.05).

## Data Availability

Not available.

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
