# Peer review of "Characterization of Effects of Different Tea Harvesting Seasons on Quality Components, Color and Sensory Quality of “Yinghong 9” and “Huangyu” Large-Leaf-Variety Black Tea"

_molecules, 2022, doi:10.3390/molecules27248720_

Round 1

Reviewer 1 Report

This work aims to understand the effect of harvesting seasons on physicochemical qualities, the quality components, color and sensory characterization of black tea produced by “Yinghong 9” (Yh) and its mutant “Huangyu” (Hy) leaves were analyzed. A topic that hasn't been well researched as of yet. The manuscript is well structured and the figures are easy to comprehend. The results are valuable for the community and worth to be published. However, the manuscript shows a variety of issues that should be addressed.

The content of the introduction was insufficient to reflect the necessity of the author's research.

Line 68: When were the fresh leaves collected?

 Line 146-148: “As to low accumulation of soluble sugar content in Hy, that was closely correlated with the reduction of photosynthesis leading to low levels of carbohydrate synthesis” , the results of the manuscript were insufficient to support this conclusion.

Line 157: The conclusion was not clear yet, and the author should refine the main results of this session.

Line: What is "wooden surfaces".

Line190-192: The conclusion was not a summary of the analysis in 3.2.

Line 199: Table 1 should be Table 2?

Line 203-205: The statement was inconsistent with the result of Table 1.

Author Response

  1. The content of the introduction was insufficient to reflect the necessity of the author's research.

RE: Thanks for your comments. We have added more information in the introduction in the revised manuscript to reflect the necessity of our research.

  1. Line 68: When were the fresh leaves collected?

RE: Thanks for your question. The fresh leaves of Yh and Hy were collected in spring: April 5, summer: July 10 and autumn:September 12, respectively.

  1. Line 146-148: “As to low accumulation of soluble sugar content in Hy, that was closely correlated with the reduction of photosynthesis leading to low levels of carbohydrate synthesis” , the results of the manuscript were insufficient to support this conclusion.

RE: Thanks for your comments. Since the results of the manuscript were insufficient to support this conclusion, we have deleted this sentence in the revised manuscript.

  1. Line 157: The conclusion was not clear yet, and the author should refine the main results of this session.

RE: Thanks for your comments. We have rewritten the conclusion in the revised manuscript.

  1. Line: What is "wooden surfaces".

RE: Thanks for your comments. “wooden surfaces” is the native wooden surfaces with equal surfaces, there are color differences (Eab) with a magnitude of 1 to 2.

  1. Line190-192: The conclusion was not a summary of the analysis in 3.2.

RE: Thanks for your comments. We have rewritten the conclusion in the revised manuscript.

  1. Line 199: Table 1 should be Table 2?

RE: Thanks for your comments, we have modified this in the revised version.

RE:  Thanks for your question.  “The different oxidation rate of NGaCs contributes to different contents of TRs and TFs” in line199 was showed in Fig 1C. So, Table 1 is OK.

  1. Line 203-205: The statement was inconsistent with the result of Table 1.

RE:  Thanks for your question. we have modified this in the revised version.

Reviewer 2 Report

Presented for review manuscript with title: “Characterization of Different Tea Harvesting Seasons on Nonvolatile Compounds, Color and Sensory Quality of “Yinghong 9” and “Huangyu” Large Leaf Variety Black Tea” is very interesting and well written.

 The manuscript authors the introduction section presented the current state of knowledge on the experimental design. The topic is a very new, because touch a problem of quality and chemical composition as well sensory properties of different teas, according to their harvesting season.

 Abstract section is present properly.

 The Introduction section includes all necessary information about examined objects and problems.

 Material and methods

Page 2, lines 68-72. How many leaves were use per one experimental combination? Please add this information to manuscript text.

 Results section are presented in clear and easily way.

 The discussion section presents a very good comparison of the obtained results with other results available in the data basis.

 General opinion: I think, that presented manuscript is a very valuable with extremely high scientific value and should be published after minor correction according my remarks in Molecules journal.

Author Response

  1. Page 2, lines 68-72. How many leaves were use per one experimental combination? Please add this information to manuscript text.

RE: Thanks for your comments. There were 25kg fresh leaves were use per one experimental combination, and we have added this information in the methods in the revised version.

  1. Results section are presented in clear and easily way.

RE: Thanks very much!

  1. The discussion section presents a very good comparison of the obtained results with other results available in the data basis.

RE: Thanks very much!

Reviewer 3 Report

In my opinion, the content of the manuscript entitled "Characterization of Different Tea Harvesting Seasons on Quality Components, Color and Sensory Quality of “Yinghong 9” and “Huangyu” Large Leaf Variety Black Tea" is interesting. The results provide a scientific basis for understanding the effects of the harvesting seasons on the flavor quality of black tea.  For this reason, the paper could run for publication after minor suggestions.

1.     Line 26 and 262: is “sPLS-DA” or “PLS-DA”, please check.

2.     In the Materials and Methods, Amino acids and soluble sugar determination is too brief, would you please add more detail. In addition, please add the detail methods of tea pigments determination.

3.     In the Results and discussion, Figure 1C shows catechins oxidation efficiency, but there is no method description in the Materials and Methods, please add method detail of oxidation efficiency.

4.     Are there any quality assurance parameters for the chemical analysis done in this study?

5.     Line 201: “Spring, Summer and Autumn” should be in lower case, please modify.

6.     Line 300: punctuation mark was missed in References 2.

7.     Line 316: references 10, 11, 24 is not the same format with others, all references should be in the same format according to the journal's requirements, please modify.

8.     In the caption of Figure 1, “the determination of” should be deleted.

9.     It would be better if the authors presented the contents of tea polyphenol, caffeine, amino acid and soluble sugar as mg/g dry weight.

Author Response

  1. Line 26 and 262: is “sPLS-DA” or “PLS-DA”, please check.

RE: Thanks for your comments. It is PLS-DA, and we have modified this in the revised version.

  1. In the Materials and Methods, Amino acids and soluble sugar determination is too brief, would you please add more detail. In addition, please add the detail methods of tea pigments determination.

RE: Thanks for your comments. We have added more detail for amino acids, tea pigments and soluble sugar determination in the modified manuscript.

  1. In the Results and discussion, Figure 1C shows catechins oxidation efficiency, but there is no method description in the Materials and Methods, please add method detail of oxidation efficiency.

RE: Thanks for your comments. The detail of catechin oxidation efficiency were added in the Materials and Methods.

  1. Are there any quality assurance parameters for the chemical analysis done in this study?

RE: Thanks for your comments. The chemicals were analyzed according to national standards, and reference materials used for analysis of the chemicals were purchased from Sigma. In addition, chemical quality test were done in the institute of fruit and tea, Hubei academy of agricultural Sciences, which have rich experiences in detecting physical and chemical quality of tea. Based on the above condition, quality assurance parameters for the chemical analysis are absolutely guaranteed.

  1. Line 201: “Spring, Summer and Autumn” should be in lower case, please modify.

RE: Thanks for your comments. We have modified in the revised manuscript.

  1. Line 300: punctuation mark was missed in References 2.

RE: Thanks for your comments. We have modified it in the revised manuscript.

  1. Line 316: references 10, 11, 24 is not the same format with others, all references should be in the same format according to the journal's requirements, please modify.

RE: Thanks for your comments. We have modified all references in the revised manuscript.

  1. In the caption of Figure 1, “the determination of” should be deleted.

RE: Thanks for your comments. We have modified it in the revised manuscript.

  1. It would be better if the authors presented the contents of tea polyphenol, caffeine, amino acid and soluble sugar as mg/g dry weight.

RE: Thanks for your comments. We have changed it as your suggestion in the revised manuscript.